# Non-canonical endogenous expression of voltage-gated sodium channel Na_V1.7 subtype by the TE671 rhabdomyosarcoma cell line

Neville M. Ngum, Muhammad Y. A. Aziz, Liaque Mohammed Latif, Richard J. Wall, Ian R. Duce 🆔 and Ian R. Mellor 🆔

*School of Life Sciences, University of Nottingham, Nottingham, UK*

Edited by: Ian Forsythe & Nikita Gamper

The peer review history is available in the Supporting information section of this article (https://doi.org/10.1113/JP283055#support-information-section).

**Abstract** The human TE671 cell line was originally used as a model of medulloblastoma but has since been reassigned as rhabdomyosarcoma. Despite the characterised endogenous expression of voltage-sensitive sodium currents in these cells, the specific voltage-gated sodium channel (VGSC) subtype underlying these currents remains unknown. To profile the VGSC subtype in undifferentiated TE671 cells, endpoint and quantitative reverse transcription–PCR (qRT-PCR), western blot and whole-cell patch clamp electrophysiology were performed. qRT-PCR profiling revealed that expression of the *SCN9A* gene was ∼215-fold greater than the *SCN4A* gene and over

**Neville Ngum** is a postdoctoral research associate with expertise in ion channel physiology and pharmacology. He obtained his PhD from the University of Nottingham in 2019 where he investigated the components of the giant Indian centipede for Na_V1.7 ion channel modifiers. He then moved to the University of East Anglia to research on novel antagonists of P2X4 receptors and he is currently exploring methods of non-invasive dynamic neural control by laser-based technology for treatments of neuro-degenerative diseases at Aston University.

400-fold greater than any of the other VGSC genes, while western blot confirmed that the dominant *SCN9A* RNA was translated to a protein with a molecular mass of ∼250 kDa. Elicited sodium currents had a mean amplitude of $2.6 \pm 0.7$ nA with activation and fast inactivation $V_{50}$ values of $-31.9 \pm 1.1$ and $-69.6 \pm 1.0$ mV, respectively. The currents were completely and reversibly blocked by tetrodotoxin at concentrations greater than 100 nM ($IC_{50} = 22.3$ nM). They were also very susceptible to the $Na_V1.7$ specific blockers Huwentoxin-IV and Protoxin-II with $IC_{50}$ values of 14.6 nM and 0.8 nM, respectively, characteristic of those previously determined for $Na_V1.7$. Combined, the results revealed the non-canonical and highly dominant expression of $Na_V1.7$ in the human TE671 rhabdomyosarcoma cell line. We show that the TE671 cell line is an easy to maintain and cost-effective model for the study of $Na_V1.7$, a major target for the development of analgesic drugs and more generally for the study of pain.

(Received 7 March 2022; accepted after revision 5 April 2022; first published online 12 April 2022)

**Corresponding author** I. R. Mellor: School of Life Sciences, University of Nottingham, University Park, Nottingham NG7 2RD, UK. Email: ian.mellor@nottingham.ac.uk

**Abstract figure legend** Phase contrast image of undifferentiated TE671 cells and the experimental approaches used to confirm dominant expression of the $Na_V1.7$ subtype of voltage-gated sodium channels. These included PCR, western blotting and patch-clamp electrophysiology with $Na_V1.7$-specific toxins such as Protoxin-II from the tarantula spider, *Thrixopelma pruriens*.

### Key points

- Undifferentiated TE671 cells produce a voltage-sensitive sodium current when depolarised.
- The voltage-gated sodium channel isoform expressed in undifferentiated TE671 cells was previously unknown.
- Through qRT-PCR, western blot and toxin pharmacology, it is shown that undifferentiated TE671 cells dominantly (>99.5%) express the $Na_V1.7$ isoform that is strongly associated with pain.
- The TE671 cell line is, therefore, a very easy to maintain and cost-effective model to study $Na_V1.7$-targeting drugs.

## Introduction

The initiation and propagation of action potentials is dependent on voltage-gated sodium channels (VGSC) expressed by central and peripheral neurons, cardiac and skeletal muscle myocytes, and neuroendocrine cells (de Lera Ruiz & Kraus, 2015). These VGSCs are transmembrane protein complexes consisting of a single $\alpha$-subunit (∼260 kDa) that forms the ion conducting pore of the channel, associated with auxiliary $\beta$-subunits (33–45 kDa): a non-covalently linked $\beta1$ or $\beta3$ and a covalently linked $\beta2$ or $\beta4$ subunit (Hull & Isom, 2018). The eukaryotic $\alpha$-subunit is a pseudo-tetramer containing four domains folded together to create a central pore whose structural constituents are responsible for the channel gating, sodium selectivity and conductance. Each of the domains contains six $\alpha$-helical transmembrane segments (S1–S6) with the segments connected either by a non-conserved small hydrophilic intracellular loop connecting the S2–S3 and S4–S5 segments, or a non-conserved small hydrophilic extracellular loop connecting S1–S2 and S3–S4 segments.

The S5–S6 segments, on the other hand, are connected by long extracellular sequences incorporating the P-loops that form a membrane re-entrant loop embedded in the transmembrane region (Guy & Seetharamulu, 1986). The four homologous domains are linked by large intracellular loops and the N- and C-termini are both placed intracellularly. The ion-conducting pore subdomain is formed by the four sets of S5 and S6 segments and their membrane re-entrant P-loops where the DEKA sodium selectivity filter is situated, while the S1–S4 segments in each domain forms the voltage-sensing subdomains of the channel (de Lera Ruiz & Kraus, 2015). Nine mammalian VGSC genes encode nine different $\alpha$-subunits defining the distinct sodium channel subtypes ($Na_V1.1$–$Na_V1.9$) that contain the reception sites for numerous drugs and toxins. The localisation of the various subtypes in excitable tissues varies, with $Na_V1.1$–1.3 and $Na_V1.6$ mostly confined to the central nervous system, $Na_V1.7$–1.9 to the peripheral nervous system, whereas the expression of $Na_V1.4$ and $Na_V1.5$ is mainly restricted to the skeletal and cardiac myocytes, respectively (Kwong & Carr, 2015).

The TE671 cell line was originally reported as being derived from a cerebellar medulloblastoma growing in culture as undifferentiated preneuroglial cells with no glial or neural elements (McAllister et al., 1977). The cell line was characterised as possessing neuronal nicotinic acetylcholine receptors (nAChR) blocked by $\alpha$-bungarotoxin (Syapin et al., 1982), but later studies with monoclonal antibodies revealed that the channel had an $(\alpha 1)_2 \beta 1 \gamma \delta$ stoichiometry, depicting an embryonic muscle-type nAChR (Luther et al., 1989). This implied a muscle rather than a neuronal origin of these cells. A rare point mutation at the third base of codon 61 in the N-*ras* gene, a trademark feature for the rhabdomyosarcoma RD cell line, was reported in TE671 cells coupled to cytogenetic studies that revealed the same marker chromosomes for the two cell lines (Stratton et al., 1989). DNA fingerprinting finally confirmed that TE671 is a derivative of the same line as RD cells and it has ever since been resolved as an RD cell line (Chen et al., 1989). However, this has not stopped some authors from placing the TE671 cell line amongst the list of cell lines with possible cross-contaminations at their initial stages of discovery (Capes-Davis et al., 2010). This cross-contamination may be particularly evident for TE671 cells as contrasting origins were reported by the same laboratory that ascribed TE671 to a medulloblastoma origin (McAllister et al., 1977) only after they had initially reported a rhabdomyosarcoma cell line derived from a pelvic sarcoma on a 7-year-old female (McAllister et al., 1969). The TE671 cell line has been extensively used as an alternative to human muscle as a source of human muscle-type nAChR (Brier et al., 2003; Franciotta et al., 1999; Makino et al., 2017; Patel et al., 2020; Somnier, 1994; Stromgaard et al., 1999). It is also known to express ATP-sensitive K$^+$ channels (Miller et al., 1999), small conductance calcium-activated SK3 channels (Carignani et al., 2002), human Eag1 potassium channels (K$_V$10.1) (Mello de Queiroz et al., 2006), and has been used as a system for studying VGSCs (Gambale & Montal, 1990). These VGSCs exhibited the characteristic features of 'classical' sodium channels of excitable cells; voltage-gating properties, cation selectivity, and tetrodotoxin sensitivity (Gambale & Montal, 1990) and have been recruited as targets to characterise scorpion toxins specific to VGSC (Barona et al., 2006).

Despite the use of TE671 cells as an endogenous source of VGSC, very limited reference is made to the exact VGSC subtype contributing the inward currents, with disagreement as to the presence of neuronal VGSC subtypes (Fakler et al., 1990; Gambale & Montal, 1990), the skeletal muscle VGSC subtype (Barrett-Jolley et al., 1994), or a mixture of various VGSC subtypes (Na$_V$ 1.1 to Na$_V$ 1.7) in culture (Barona et al., 2006). To this end, we have deciphered the VGSC subtype expressed by the rhabdomyosarcoma TE671 cell line in its undifferentiated form through gene expression and pharmacological dissection. We have provided quantitative evidence for the subtype responsible for the inward current, assessed its sensitivity to three VGSC blockers and characterised its electrophysiological properties. Subtype characterisation of VGSC would make this cell line a more attractive tool for specific subtype research.

## Methods

### Materials

Primers, all cell culture reagents and tetrodotoxin (TTX) were from Sigma-Aldrich. Huwentoxin-IV (HWTX-IV) and Protoxin-II (ProTx-II) were from Tocris Bioscience, UK.

### TE671 cell culture

TE671 cells (RRID:CVCL_1756) were obtained from the European Collection of Authenticated Cell Cultures (ECACC; catalogue no. 89071904) and were grown as monolayer cultures (25 cm$^2$ flasks) in Dulbecco's modified Eagle's medium (DMEM) containing 10% (v/v) fetal bovine serum (FBS), 2 mM glutamine, 10 IU/ml penicillin and 10 $\mu$g/ml streptomycin at 37°C in a humidified 5% CO$_2$–95% air environment. Cell stocks were stored in liquid nitrogen in medium containing 65% DMEM, 25% FBS and 10% dimethyl sulfoxide and, following recovery, were passaged once per week up to passage 20 after which a new batch of cells was recovered from liquid nitrogen stocks. Cells were passaged by treatment with 2.5% trypsin/EDTA and were re-plated at 40,000 cells/cm$^2$. For electrophysiological experiments, cells were plated onto heat-sterilised cut sections (about 5 × 20 mm) of glass coverslips in 35 mm Petri dishes and incubated for at least 48 h before experiments.

### Isolation of RNA and polymerase chain reaction

RNA was extracted from five replicates of 1 × 10$^6$ cell samples after 5, 10 and 18 passages in continuous culture using the RNeasy Plus Mini Kit (Qiagen, Germantown, MD, USA) as per the manufacturer's instructions. cDNA strands were synthesised from 0.4 $\mu$g total extracted RNA using a qPCRBIO cDNA Synthesis Kit (PCR Biosystems, London, UK) and used as templates for polymerase chain reaction amplification via the G-Storm GS1 Thermal Cycler (Gene Technologies Ltd, London, UK). Forward and reverse oligonucleotide primers of 19−23 bp in length were designed with PrimerBank online software (https://pga.mgh.harvard.edu/primerbank/) with 40−60% GC content. A set of two primers, designed using Primer 3 version 3 (Primer-blast, NCBI) (Table 1), were used

**Table 1. Details of primer sequences used for the PCR experiments**

| Primer name | Primer bank ID | Sequence | Product (bp) | Description |
|---|---|---|---|---|
| P1 | 320461721c1 | TCTCTTGCGGCTATTGAAAGAC | 86 | *SCN1A* forward 61–82 |
| P2 | | GGGCCATTTTCGTCGTCATCT | | *SCN1A* reverse 146–126 |
| P3 | 320461721c3 | ATGTGGAAATAGCTCTGATGCAG | 109 | *SCN1A* forward 1005–1027 |
| P4 | | AGCCCAACTGAAGGTATCAAAG | | *SCN1A* reverse 1113–1092 |
| P5 | 93141213c1 | CGCTTCTTTACCAGGGAATCC | 77 | *SCN2A* forward 43–63 |
| P6 | | TCCTGTTTGGGTCTCTTAGCTTT | | *SCN2A* reverse 119–97 |
| P7 | 93141213c3 | TCTAAGCGTGTTTGCGCTAAT | 139 | *SCN2A* forward 774–794 |
| P8 | | ACCATTCCCATCCAATGAATTGT | | *SCN2A* reverse 912–890 |
| P9 | 126362946c1 | GGAGAGCTGTTGGAAAGTTCTT | 77 | *SCN3A* forward 1435–1456 |
| P10 | | TTCCTTCGGTTCCTCCATTCT | | *SCN3A* reverse 1511–1491 |
| P11 | 126362946c3 | CTCAGAAACCCATACCTCGCC | 169 | *SCN3A* forward 4364–4384 |
| P12 | | CGGGACAAAACTAGGGTCATGTA | | *SCN3A* reverse 4532–4510 |
| P13 | 93587341c2 | CATCGTACTCAACAAGGGCAA | 87 | *SCN4A* forward 279–299 |
| P14 | | CGCCTGACTACGCTGAAGG | | *SCN4A* reverse 365–347 |
| P15 | 93587341c3 | TTCACAGGGATCTACACCTTTGA | 141 | *SCN4A* forward 490–512 |
| P16 | | CACAAACTCTGTCAGGTACGC | | *SCN4A* reverse 630–610 |
| P17 | 237512981c1 | TCTCTATGGCAATCCACCCCA | 145 | *SCN5A* forward 198–218 |
| P18 | | GAGGACATACAAGGCGTTGGT | | *SCN5A* reverse 342–322 |
| P19 | 237512981c3 | AGCTGGCTGATGTGATGGTC | 94 | *SCN5A* forward 746–765 |
| P20 | | CACTTGTGCCTTAGGTTGCC | | *SCN5A* reverse 839–820 |
| P21 | 374429548c1 | CCTTTCACCCCTGAGTCACTG | 131 | *SCN8A* forward 46–66 |
| P22 | | AGGTCGCTGTTTGGCTTGG | | *SCN8A* reverse 176–158 |
| P23 | 374429548c2 | CAAACAGCGACCTGGAAGCA | 111 | *SCN8A* forward 164–183 |
| P24 | | TCTGCGTCAAATAGTATGGGTCA | | *SCN8A* reverse 274–252 |
| P25 | 256017191c2 | AGAGGGGTACACCTGTGTGAA | 192 | *SCN9A* forward 975–995 |
| P26 | | CCCAGGAAAATCACTACGACAAA | | *SCN9A* reverse 1166–1144 |
| P27 | 256017191c3 | ATTCGTGGCTCCTTGTTTTCTG | 206 | *SCN9A* forward 1615–1636 |
| P28 | | CTACTGGCTTGGCTGATGTTAC | | *SCN9A* reverse 1820–1799 |
| P29 | 110835709c1 | TCCCTCGAAACTAACAACTTCCG | 115 | *SCN10A* forward 19–41 |
| P30 | | TCTGCTCCCTATGCTTCTCTC | | *SCN10A* reverse 133–113 |
| P31 | 110835709c3 | TTCCCGGTTTAGTGCCACTC | 79 | *SCN10A* forward 303–322 |
| P32 | | AGACACTTTGATGGCCGTTCT | | *SCN10A* reverse 381–361 |
| P33 | 115583666c1 | CCCTTCACTTCCGACTCTCTG | 99 | *SCN11A* forward 52–72 |
| P34 | | AGGCTGGGGTACTTCTCCTG | | *SCN11A* reverse 150–131 |
| P35 | 115583666c2 | CCCAAGCTCTATGGCGACATT | 126 | *SCN11A* forward 187–207 |
| P36 | | ACTGAAGCGGTAGATTGTCCTC | | *SCN11A* reverse 312–291 |
| P37 | 378404907c1 | GGAGCGAGATCCCTCCAAAAT | 197 | *GAPDH* forward |
| P38 | | GGCTGTTGTCATACTTCTCATGG | | *GAPDH* reverse |

for each channel subtype and *GAPDH* was employed as the reference gene. A complete master mix for PCR was prepared using Q5 High-Fidelity DNA Polymerase (New England Biolabs Inc., Hitchin, UK) with the following cycling conditions; 98°C for 30 s for denaturation, followed by 30 cycles of amplification (98°C for 10 s, 65–69°C for 30 s, 72°C for 30 s) and a final 72°C for 2 min for extension. PCR products were analysed by 2% agarose gel electrophoresis and visualised with ethidium bromide staining. The amplified DNA genes were sequenced using 3130xl ABI PRISM Genetic Analyzer (Thermo Fisher Scientific, Waltham, MA, USA) and BigDye version 3.1 (Thermo Fisher Scientific). The SYBR Green QPCR master mix (Thermo Fisher Scientific) was used for quantitative reverse transcription–PCR (qRT-PCR) and analysis was conducted using an Applied Biosystems 7500 fast machine (Thermo Fisher Scientific) with the following cycling conditions: 95°C for 20 s followed by 40 cycles of 95°C for 3 s; 60°C for 30 s. Relative gene expression was determined using the comparative cycle threshold method (Schmittgen & Livak, 2008).

## Western blotting

Undifferentiated TE671 cells at passages 12 and 13 were grown to confluence in a 75 cm$^2$ flask as described above ('TE671 cell culture'). The culture medium was

aspirated and cells were washed in, then incubated in, $Ca^{2+}$- and $Mg^{2+}$-free phosphate-buffered saline for 5 min before being agitated and transferred to a centrifuge tube. The cell sample was centrifuged at 93$g$ for 10 min to pellet the cells and the supernatant removed. Two hundred microlitres of lysis buffer (20 mM Tris–HCl, 1 mM EGTA, 320 mM sucrose, 0.1% Triton X-100, 1 mM NaF, 10 mM $\beta$-glycerophosphate, pH 7.6) was added to the pellet and the cells resuspended using a pellet pestle. The resuspended cells were added to an Eppendorf tube and placed on ice for 10 min, inverting the tube every minute. The sample was then centrifuged at 14549$g$ for 10 min at 4°C and 150 $\mu$l of the supernatant removed and added to 150 $\mu$l 2× solubilisation buffer (125 mM Tris-HCl, 20% glycerol, 2% SDS, 10% $\beta$-mercaptoethanol, 0.1% bromo-phenol blue), generating a 1:2 dilution of protein. Protein samples were denatured by placing in a heating block at 95°C for 5 min followed by vortexing then centrifuging at 14549$g$ for 1 min before loading onto a polyacrylamide gel alongside Precision Plus Protein Dual Colour Standards (Bio-Rad Laboratories, Watford, UK). The gel was run in electrophoresis buffer (25 mM Tris–HCl, 192 mM glycine, 0.1% SDS) at 200 V for 45 min. Proteins were transferred from the gel to a nitrocellulose membrane in transfer buffer (25 mM Tris–HCl, 192 mM glycine, 20% methanol) at 100 V for 60 min. Transfer was confirmed by adding 0.5% Ponceau stain to the nitrocellulose membrane. The membrane was rinsed in TBST (25 mM Tris–HCl, 125 mM NaCl, 0.1% Tween 20, pH 7.6) and then placed in 5% dried skimmed milk in TBST for 1 h with rocking. The membrane was then incubated overnight at room temperature in the Na$_V$1.7 primary rabbit antibody (Cell Signaling Technology, Danvers, MA, USA, cat. no. 14573; Alvarez et al., 2021) diluted 1:500 in 5% dried skimmed milk/TBST, with rocking. The membrane was rinsed three times in TBST followed by three 5-min and three 15-min washes in TBST, then incubated for 1 h at room temperature in the secondary antibody (IRDye 800CW goat anti-rabbit IgG; LI-COR Biosciences, Cambridge, UK) diluted 1:10 000 in 5% dried skimmed milk/TBST, followed by the same wash sequence as after the primary antibody. Finally, the nitrocellulose membrane was imaged using a LI-COR Scanner with Image Studio V5.2 software.

## Patch-clamp electrophysiology measurements

Whole-cell patch-clamp electrophysiology was performed using an Axopatch 200A (Axon instruments, San Jose, CA, USA) patch-clamp amplifier and currents recorded to a PC using WinWCP V4.5.7 software (Dr John Dempster, University of Strathclyde, UK). Patch-pipettes were pulled from borosilicate glass capillaries (World Precision Instruments, Sarasota, FL, USA, 1B150F-4) using a P-97 Flaming–Brown micropipette puller (Sutter Instrument Co., Novato, CA, USA) to give a resistance of ∼3–5 MΩ when filled with a caesium intracellular solution (in mM): 140 CsCl, 10 NaCl, 1 CaCl$_2$, 1 MgCl$_2$, 11 EGTA and 5 Hepes (pH 7.2 with CsOH). Cells were transferred to a perfusion chamber mounted on the stage of an inverted microscope and perfused with an external solution containing (in mM): 135 NaCl, 5.4 KCl, 1 CaCl$_2$, 1 MgCl$_2$, 5 Hepes, 10 D-glucose (pH 7.4 with NaOH). Cells were held at a resting potential of −80 mV for 4 min to allow equilibration between pipette solution and intracellular solution before recordings were initiated. Series resistance errors were compensated with 75−90% series resistance compensations in the amplifier. Analog signals were filtered at 10 kHz using a lowpass Bessel filter and leak currents subtracted using the P/4 method. Signals were sampled at 8–41 kHz and digitised using a PCI-6221/BNC-2110 digital interface (National Instruments, Austin, TX, USA). Stock solutions of TTX, HWTX-IV and ProTx-II were diluted in the bath solution to desired concentrations and applied using a DAD-12 superfusion system (Adams and List Associates, Westbury, NY, USA). The latter two peptide toxins from spiders were chosen for pharmacological characterisation as the two most selective Na$_V$1.7 inhibitors known, compared to, for example, the small molecule inhibitors such as PF-05089771.

## Voltage protocols

To study voltage dependence of activation and rate of fast inactivation of VGSCs, TE671 cells were clamped at −80 mV then subjected to a series of 25 ms step depolarisation test pulse potentials from −70 to +60 mV in 10 mV increments at 1 s intervals. Use-dependent inhibition of elicited Na$^+$ currents by various toxins was assessed through a series of step depolarisations from −80 mV to −10 mV for 25 ms applied every 5 s over 4 min. Voltage dependence of steady-state fast inactivation was studied by clamping cells at a holding potential of −80 mV before application of 200 ms inactivation pre-pulse potentials ranging from −130 to +10 mV in 10 mV increments followed by a 10 ms test step to −10 mV for 25 ms before returning to the holding potential (−80 mV). To study voltage dependence of steady state slow inactivation, cells were clamped to a holding potential of −80 mV before application of long duration pre-pulses of 10 s at voltages ranging from −130 to +10 mV, in 10 mV increments. This was followed by a repolarising step to −80 mV for 10 ms to enable recovery from fast inactivation, before a 25 ms test-pulse to −10 mV was applied. To study the time dependence of recovery from fast inactivation, cells were clamped at holding potentials of −100, −90, −80 or −70 mV

followed by application of a first depolarisation step to −10 mV for 10 ms to evoke Na$^+$ currents and fast inactivation before immediately returning to the pre-pulse holding potential for durations of 10–300 ms (in 10 ms increments) to allow recovery from inactivation. This was followed by a second depolarisation test pulse to −10 mV to evoke Na$^+$ currents that were then compared with the amplitude of the current obtained in response to the first depolarising step.

## Data analyses

Data analyses including data transforms, graph plotting, curve fitting and statistical tests were performed using GraphPad Prism 8 software (GraphPad Software, Inc., La Jolla, CA, USA). Current amplitudes and conductance were expressed as a means ± standard error of the mean (SEM).

Following the activation protocol, the current amplitudes recorded at each voltage step were plotted against the test potential and fitted with eqn (1) to produce current–voltage (*I–V*) relationships and to yield the reversal potential ($V_{rev}$) and $G_{max}$ for each cell:

$$I_{peak} = \frac{G_{max}\,(V_{T.act} - V_{rev})}{1 + \exp\left(\frac{V_{50.act} - V_{T.act}}{k}\right)} \qquad (1)$$

Using Ohm's law, peak current data for each test potential ($V_{T.act}$) was then converted to the corresponding conductance using eqn (2):

$$G = \frac{I_{peak}}{V_{T.act} - V_{rev}} \qquad (2)$$

where $G$ is the conductance, $I_{peak}$ is the peak current evoked by the depolarising test potential ($V_{T.act}$) and $V_{rev}$ is the reversal potential.

The relative conductances were normalised to $G_{max}$, plotted as a function of applied test potentials and fitted with a Boltzmann sigmoidal equation (eqn (3)) to yield the half-maximal activation voltage (the voltage at which half the number of available channels become activated) ($V_{50.act}$):

$$\frac{G}{G_{max}} = Min + \frac{Max - Min}{1 + \exp\left(\frac{V_{50.act} - V_{T.act}}{k}\right)} \qquad (3)$$

where $G/G_{max}$ is normalised conductance at the depolarising step test potential ($V_{T.act}$), and the normalised conductance varies from a minimum value (Min) to a maximal value (Max), $V_{50.act}$ is the half-activation potential and $k$ is the slope factor.

The rate of fast channel inactivation was estimated by fitting single exponentials (eqn (4)) to the decaying phase of the inward sodium currents attained through the various depolarisation steps:

$$I(t) = a_0 + a_1 \exp\left(\frac{-t}{\tau_{decay}}\right) \qquad (4)$$

where $I$ is current as a function of time $t$, $a_0$ defines the final amplitude to which the current decays, $a_1$ is the peak amplitude at the start of the decay curve, and $\tau_{decay}$ represents the time constant of the decay, defining how 'fast' the curve approaches $a_0$.

Following the steady-state fast and slow inactivation protocols, the resulting currents evoked by the test pulse were normalised against the maximum and plotted as a function of each corresponding inactivating pre-pulse potential ($V_{T.inact}$) and fitted with a Boltzmann sigmoidal equation (eqn (5)) to yield the half-inactivation voltage (voltage at which half the number of available channels become inactivated) ($V_{50.inact}$). Values for both steady state fast inactivation ($V_{50.inact-f}$) and steady state slow inactivation ($V_{50.inact-s}$) were estimated:

$$\frac{I_{peak}}{I_{max}} = Min + \frac{Max - Min}{1 + \exp\left(\frac{V_{50.inact} - V_{T.inact}}{k}\right)} \qquad (5)$$

where $I_{peak}/I_{max}$ is normalised current given as a function of inactivating pre-pulse potentials ($V_{T.inact}$) varying from a minimum value to a maximum value, and $k$ is the slope factor.

For toxin concentration–inhibition curves the normalised current measurements (percentage of control response) were plotted against log of toxin concentration, and regression curves were fitted to these points using the four-parameter logistic (4PL) equation to obtain IC$_{50}$ values:

$$\% \text{ control response} = \frac{100}{1 + 10^{((\log_{10} IC_{50} - \chi) \times Hillslope)}} \qquad (6)$$

where $\chi$ is $\log_{10}$ [test compound] and IC$_{50}$ is the concentration of inhibitor causing half maximal inhibition.

Statistical comparisons were made using Student's *t*-test. Data distribution was tested using both D'Agostino–Pearson and Shapiro–Wilk tests for normality. A *P*-value less than 0.05 for all statistical analyses indicated a significant difference.

## Results

### TE671 cells express the *SCN9A* gene

Agarose gel results revealed good integrity of extracted RNA that was consistent between three different passages and with 28S rRNA:18S rRNA ratios of more than 1. Similarly, the OD$_{260/280}$ ratios were >1.8 while OD$_{260/230}$ ratios were >2.0, implying that RNA samples were

free of protein, and from phenolic and polysaccharide contamination. Gel bands of PCR products were observed only for the *SCN9A* gene and were consistent when both sets of primer pairs were used across three different passages (5, 10 and 18) of cells (Fig. 1*A*–*C*). There was also an apparent increase in band intensity with increasing passage number. There were no unexpected non-specific bands. The relative gene expression pattern based on qRT-PCR revealed that *SCN9A* was expressed >200-fold more than *SCN4A* and >400-fold more than any of the other VGSC genes (all *P* < 0.0001) (Fig. 1*D*). In effect, the only biologically relevant level of RNA is of that coding for Na$_V$1.7.

## Western blotting

The western blot revealed a band with a molecular mass of approximately 250 kDa (Fig. 2) as would be expected for Na$_V$1.7 and agrees well with the primary antibody manufacturer's data. The western blot was repeated twice at cell passages 12 and 13 producing the same result.

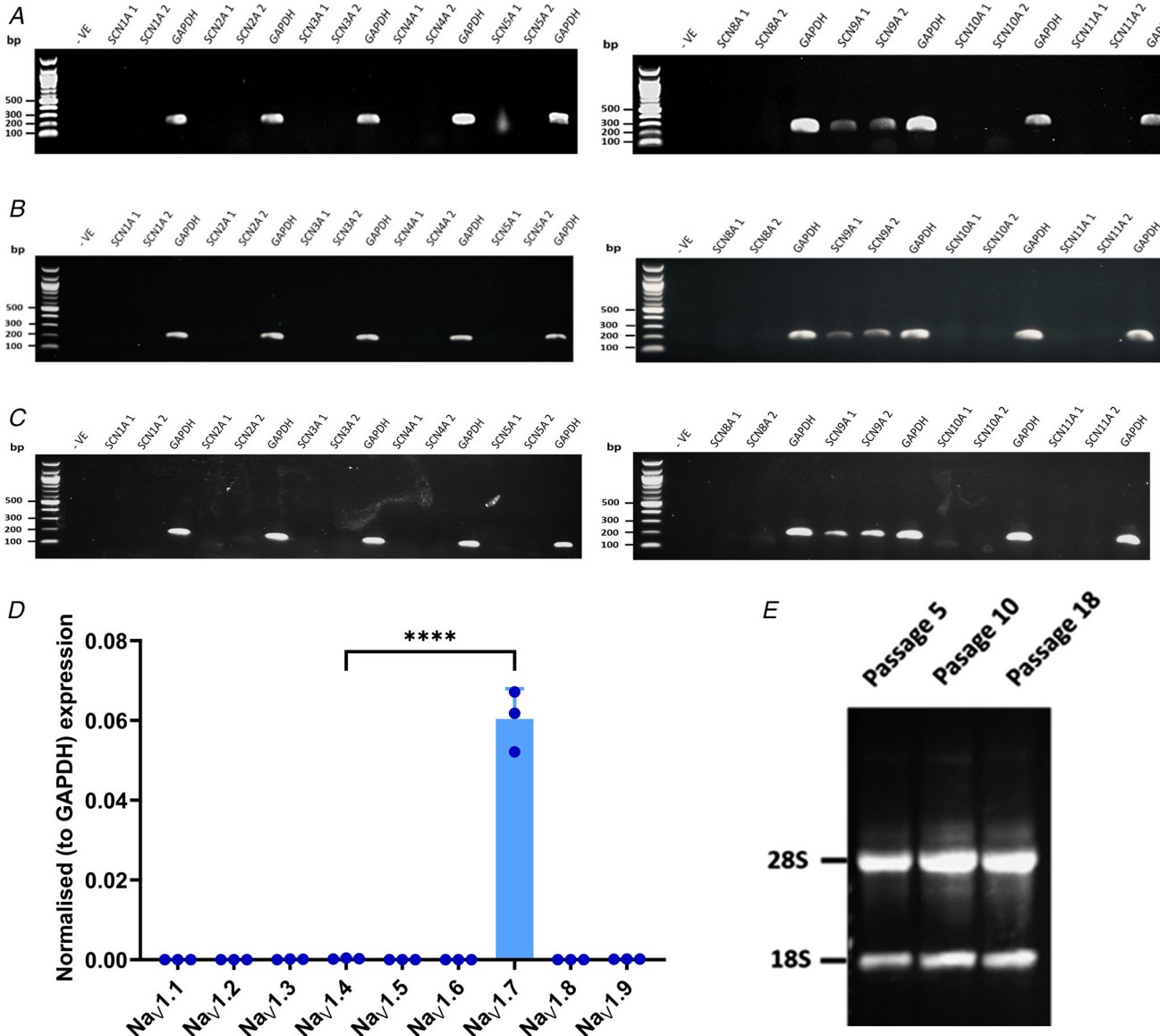

**Figure 1. Expression levels of different VGSC α-subunit isoform RNA in TE671 cells**
RNA expression of VGSC α-subunit isoform RNA at passage 5 (*A*), passage 10 (*B*), and passage 18 (*C*) using the primers given in Table 1. Product sizes are 192, 206 and 197 bp for *SCN9A* 1, *SCN9A* 2 and *GAPDH*, respectively; *n* = 5. *D*, normalised expression (means ± SD, *n* = 3 biological replicates each × 2 technical replicates) of the nine VGSC genes at passage 10 relative to *GAPDH*. ****P* < 0.0001. *E*, total RNA extraction from three different passages. [Colour figure can be viewed at wileyonlinelibrary.com]

## Properties of the sodium current endogenously expressed by TE671 cell line

A series of TE671 cell sodium currents responding to 25 ms test pulse potentials ranging from −70 to +60 mV in 10 mV increments from a −80 mV holding potential is illustrated in Fig. 3*A*. The mean maximum sodium current recorded in 25 cells was 2.6 (0.7) nA with a mean rise time of 0.75 (0.015) ms. Maximum currents occurred either at −20 or −10 mV depolarisation steps with a majority occurring at the latter. The peak inward current at each applied test voltage was normalised (fraction of maximum peak current in a series) to produce a peak current–voltage relationship (Fig. 3*B*). From this relationship, inward sodium currents were observed to be activated at test potentials more positive than −50 mV and reversed at +72.05 ± 2.89 mV, which is in accordance with the predicted Nernst equilibrium potential for Na$^+$ current under these experimental conditions. The time constant for current decay due to fast inactivation ($\tau_{\mathrm{decay}}$) at −40 mV was 4.97 (2.71) ms, decreasing to 1.05 (0.43) ms at +60 mV ($n = 11$) (Fig. 3*C*). The $V_{50.\mathrm{act}}$ obtained from the conductance–voltage relationship (Fig. 2*D*) was −31.89 ± 1.12 mV, while $V_{50.\mathrm{inact}}$ values were −69.59 ± 1.03 mV for steady state fast inactivation and −68.83 ± 0.88 mV for steady state slow inactivation (Fig. 2*D*). The time constant of recovery from fast inactivation was dependent on the repolarisation potential, decreasing from 53.9 ms at −70 mV to 9.2 ms at −100 mV (Fig. 3*E*).

## Pharmacological characterisation of TE671 voltage-gated sodium currents

The TE671 sodium currents were subjected to three different inhibitors for further subtype characterisation: TTX with selectivity for Na$_V$1.1-4 and Na$_V$1.6-7, HWTX-IV with good selectivity for Na$_V$1.7 and ProTx-II

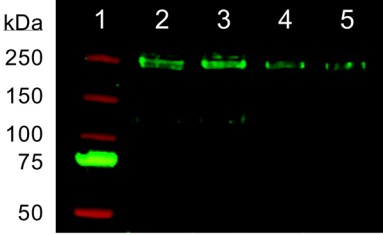

**Figure 2. Western blot demonstrating the expression of Na$_V$1.7 protein in TE671 cells**
The primary antibody was rabbit anti-Na$_V$1.7 and the secondary antibody was goat anti-rabbit IgG. Lane 1: Precision Plus Protein Dual Colour Standards; lanes 2−3: 20 $\mu$l 1:2 dilution of TE671 cell protein; lanes 4−5: 15 $\mu$l 1:2 dilution of TE671 cell protein. The band in lanes 2−5 is at ~250 kDa as expected for Na$_V$1.7. [Colour figure can be viewed at wileyonlinelibrary.com]

with excellent selectivity for Na$_V$1.7. Currents were sensitive to TTX and were almost completely blocked at concentrations higher than 100 nм. Figure 4*A* and *B* shows traces and the time course of inhibition in response to a series of −10 mV depolarisation steps before TTX application and 2 min after application of different concentrations of TTX. The current inhibition by TTX was partially reversible 2 min after the cells were washed with bath solution to 75 (7)% of the initial control response prior to TTX application (Fig. 4*B*). An IC$_{50}$ value of 22.3 nм (95% CI: 17.8–27.6 nм) was obtained from the concentration–inhibition curve (Fig. 4*C*). Like TTX, the spider toxin HWTX-IV also blocked TE671 cell VGSCs in a concentration-dependent manner (Fig. 5*A*). The time course of current evoked by −10 mV pulses at 5-s intervals and blocked with 100 nм HWTX-IV is shown in Fig. 5*B* reaching an average of 79 (6)% blockage after approximately 1 min of perfusion. The concentration–inhibition curve yielded an IC$_{50}$ value of 14.6 nм (95% CI: 10.4 – 20.2 nм) (Fig. 5*C*). Lastly, the TE671 cell sodium currents were screened against ProTx-II. Currents evoked by −10 mV pulses were highly sensitive to ProTx-II (Fig. 6*A*), with 84 (4)% inhibition achievable by 10 nм ProTx-II after about 3 min of toxin perfusion (Fig. 6*B*). The current block by ProTx-II was also concentration dependent, yielding an IC$_{50}$ value of 0.8 nм (95% CI: 0.6–10.0 nм) (Fig. 6*C*).

## Discussion

In this study, we have been able to determine the molecular profile as well as the pharmacological and electrophysiological properties of the endogenous VGSCs that contribute to the inward sodium currents in TE671 cells. Our use of molecular biology techniques for VGSC RNA detection and identification is pioneering for this cell line. The *SCN9A* gene that encodes the Na$_V$1.7 $\alpha$-subunit subtype was detected across three different cell passages (Fig. 1*A–C*). Surprisingly, no *SCN4A* gene transcription for the skeletal muscle Na$_V$1.4 was detected as one would expect for a rhabdomyosarcoma cell line. To further confirm our molecular data, end-point RT-qPCR showed a dominant expression level (~215-fold) of the Na$_V$1.7 over the Na$_V$1.4 gene (Fig. 1*D*). Transcription of genes for all the other VGSC subtypes was not detected, which disagrees with the presumption made by (Barona et al., 2006) that the TE671 cell line endogenously expresses a cocktail of Na$_V$1.1–Na$_V$1.7. Regardless of the discrepancy of the origin of TE671 cells, it is surprising that these cells would predominantly express the Na$_V$1.7 gene, thought to be localised to dorsal root ganglion (DRG) neurons, sympathetic neurons, Schwann and neuroendocrine cells (Catterall et al., 2005; Ogata & Ohishi, 2002; Savio-Galimberti et al., 2012). Our western

blot confirmed that this dominant transcription of RNA encoding $Na_V1.7$ is translated to protein of the expected molecular mass for $Na_V1.7$.

The evoked TE671 VGSC currents were voltage-dependent and maintained the typical kinetics of a sodium current response: a sigmoidal activation to peak response followed by exponential current decays (Fig. 3). As it is established that the properties of sodium currents can depend on the cells in which they are expressed (Cummins et al., 2001), it is hard to draw conclusions on the sodium channel isoform present based solely on its electrophysiological properties. For example,

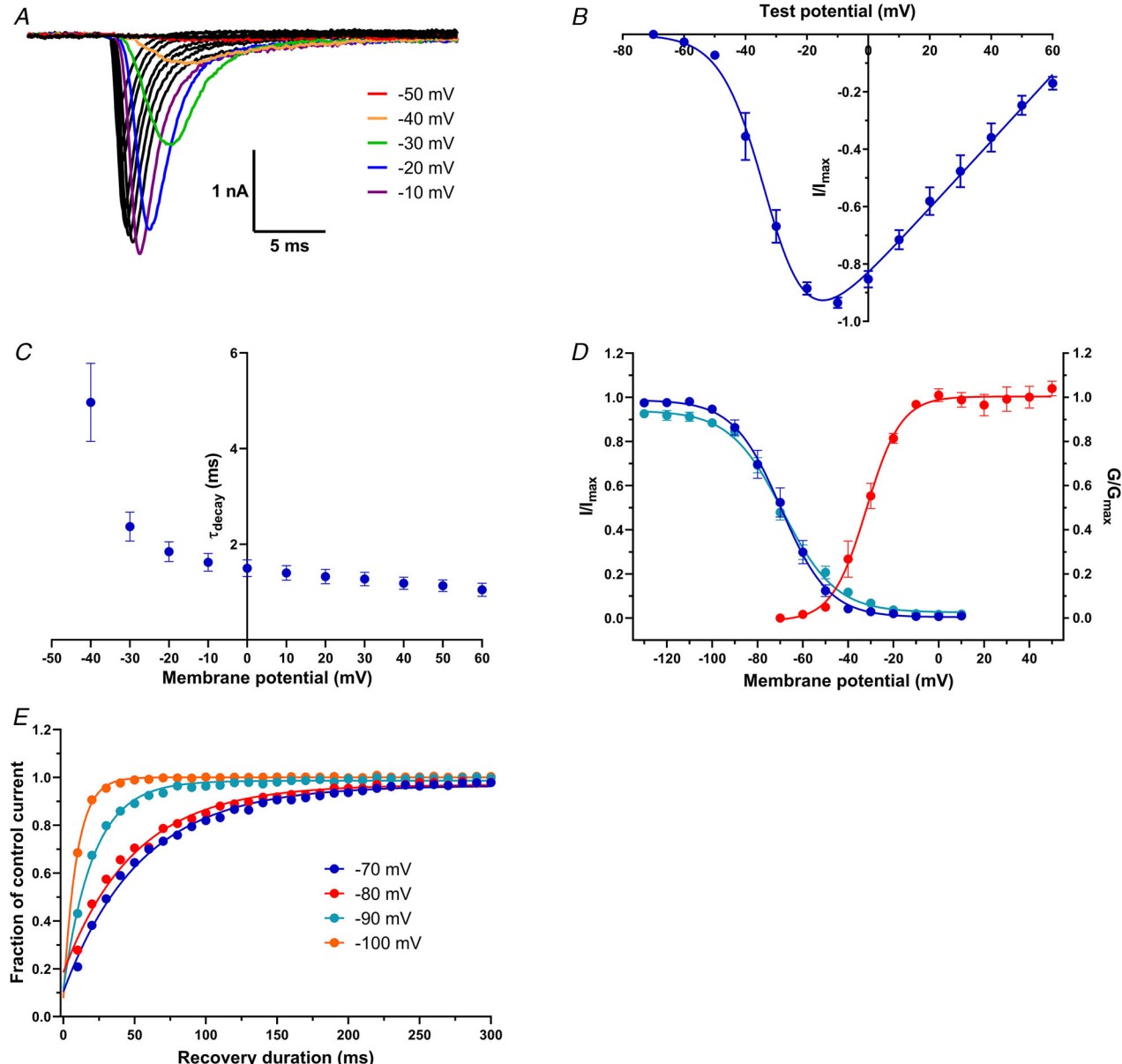

**Figure 3. Properties of the voltage-gated sodium currents elicited in TE671 cells**

*A*, family of inward currents in response to 25 ms depolarising test potentials from −70 to +60 mV in 10 mV increments. The holding potential was −80 mV. Currents in response to −50, −40 −30, −20 and −10 mV are highlighted in red, orange, green, blue and purple, respectively. *B*, normalised peak current–voltage relationship for test potentials ranging from −70 to +60 mV. Points are mean fraction of maximum current ± SEM fitted by eqn (1) (*n* = 25). *C*, mean current decay time constants ± SEM plotted as a function of membrane potential (*n* = 11). *D*, conductance–voltage relationship for activation (red circles, *n* = 25), voltage dependence of the fast steady-state inactivation (blue circles, *n* = 16) and voltage dependence of the slow steady-state inactivation (turquoise circles, *n* = 26). Points are mean fractional conductance or current ± SEM and fitted by eqn (3) or (5). *E*, time dependence of recovery from fast inactivation upon repolarisation to −70, −80, −90 or −100 mV. Points are mean fraction of control current; *n* > 20 for each data point. [Colour figure can be viewed at wileyonlinelibrary.com]

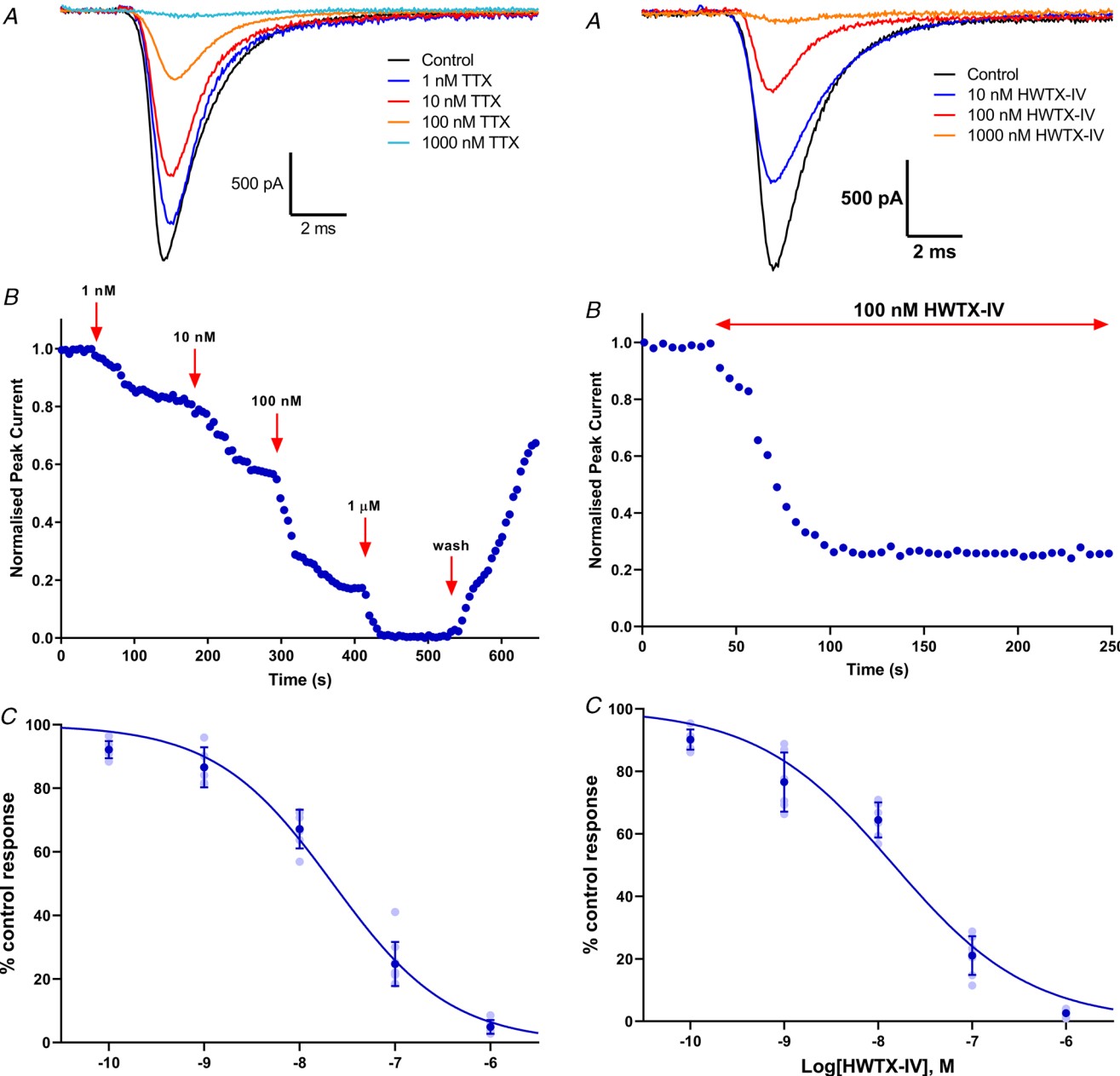

**Figure 4. Block of sodium current in TE671 cells by TTX**
*A*, representative current traces in response to depolarising steps from −80 to −10 mV, showing the inhibition of VGSC by TTX (1–1000 nM) applied for 3 min. *B*, time course of inhibition of sodium currents by various concentrations and current re-emergence following wash off. Currents in response to a depolarisation from −80 to −10 mV every 5 s were normalised to the control and plotted against time. *C*, concentration–inhibition curve for TTX block of sodium currents in TE671 cells yielding an IC$_{50}$ value of 22.3 nM (95% CI: 17.8–27.6 nM), Hill slope = −0.70 ± 0.05. Dark blue points are mean percentage of control response ± SD, $n \geq 5$ for each point, with individual data points in light blue. [Colour figure can be viewed at wileyonlinelibrary.com]

**Figure 5. Block of sodium currents in TE671 cells by HWTX-IV**
*A*, representative traces in response to depolarising steps from −80 to −10 mV, showing inhibition of VGSCs by HWTX-IV (10–1000 nM) applied for 3 min. *B*, time dependence of inhibition by 100 nM HWTX-IV of elicited sodium currents in TE671 cells. Points are normalised amplitude of the current in response to a depolarisation from −80 to −10 mV applied every 5 s. HWTX-IV was applied after 40 s. *C*, concentration–inhibition curve for HWTX-IV block of sodium currents in TE671 cells yielding an IC$_{50}$ value of 14.6 nM, Hill slope = −0.59 ± 0.05. Dark blue points are mean percentage of control response ± SD, $n \geq 5$ for each point, with individual data points in light blue. [Colour figure can be viewed at wileyonlinelibrary.com]

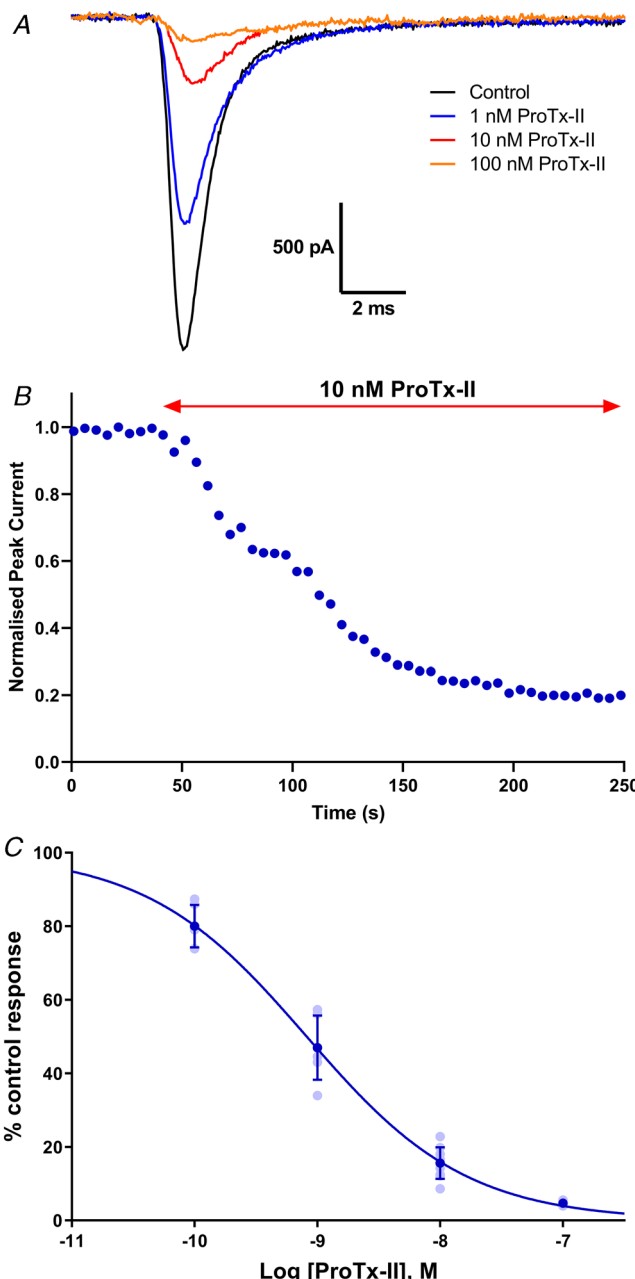

**Figure 6. Block of sodium currents in TE671 cells by ProTx-II**
*A*, representative traces in response to depolarising steps from −80 to −10 mV, showing inhibition of VGSCs by ProTx-II (1–100 nM) applied for 3 min. *B*, time-dependent inhibition by 10 nM ProTx-II on elicited sodium currents in TE671 cells. Points are normalised amplitude of the current in response to a depolarisation from −80 to −10 mV applied every 5 s. ProTx-II was applied after 40 s. *C*, concentration–inhibition curve for ProTx-II inhibition of sodium currents in TE671 cells yielding an IC$_{50}$ value of 0.8 nM, Hill slope = −0.66 ± 0.04. Dark blue points are mean percentage of control response ± SD, $n \geq 6$ for each point, with individual data points in light blue. [Colour figure can be viewed at wileyonlinelibrary.com]

while the recorded half-activation voltage is similar to those reported for wild-type Na$_V$1.7 expressed in both HEK cells (Eberhardt et al., 2014; Zeng et al., 2018) and DRG neurons (Herzog et al., 2003), it is also similar to that of Na$_V$1.4 expressed in CHO cells (Bennett, 2004). However, even though these values may fit with those obtained in our studies, a gamut of values for $V_{50act}$, such as −11.6 ± 1.1 mV for Na$_V$1.7 (Theile et al., 2011) and −19.5 ± 0.2 mV for Na$_V$1.4 (Mannikko et al., 2018), have also been reported. This pattern of variation is also consistent with values recorded for $V_{50inact-f}$ across the two channel subtypes. Exploring the repriming kinetics of these channels revealed that they recovered more rapidly from fast inactivation at more negative voltage (Fig. 3*E*), and this is consistent with reports that voltage-gated Na$^+$ channels recover from inactivation faster at more hyperpolarised voltages (Bezanilla & Armstrong, 1977). However, the recovery kinetics recorded in the present study are faster than others for Na$_V$1.7 recorded in HEK cells (Cummins et al., 1998; Han et al., 2006) or mouse DRG (Herzog et al., 2003).

The results of the mRNA detection were further supported by probing pharmacological properties of the different VGSC subtype proteins using generic and specific blockers of Na$_V$1.7 channels. The currents were sensitive to inhibition by TTX, a potent and selective guanidine-based neurotoxin that occludes the outer vestibule of the ion conducting pore (Fozzard & Lipkind, 2010). The complete blockage of TE671 cell VGSC currents by TTX corroborates the results of Gambale & Montal (1990) and implies that the voltage-sensitive currents in TE671 cells are mediated by sodium rather than calcium channels. However, a study by Fakler and colleagues revealed a varying proportion of both TTX-sensitive and TTX-insensitive sodium channels among these cells and found that while some individual cells expressed only TTX-sensitive channels, others expressed only the TTX-insensitive channels (Fakler et al., 1990). In this study, Fakler and colleagues utilised TE671 cells at an early undifferentiated growth stage characterised by large, flat and multinucleated cells. In the present study, cells at the undifferentiated growth stage displaying either long, centrinuclear bipolar spindle or flat monopolar mononuclear shapes were used, and their VGSC currents were completely blocked by 500 nM TTX. This therefore implied a TTX sensitive VGSC subtype (Na$_V$1.1–1.4, Na$_V$1.6 and/or Na$_V$1.7), as opposed to the higher micromolar range demonstrated for TTX-insensitive VGSCs (Na$_V$1.5, Na$_V$1.8 and Na$_V$1.9) (Catterall et al., 2005). The TTX IC$_{50}$ value recorded in this study is similar to the 24.5 nM (Klugbauer et al., 1995) and 36 ± 7 nM (Tsukamoto et al., 2017) values reported for human Na$_V$1.7, and 24.5 nM reported for human Na$_V$1.4 (Chahine et al., 1994).

The TE671 cell VGSCs were further subjected to HWTX-IV, a gating modifier neurotoxin from the *Ornithoctonus huwena* tarantula, which preferentially inhibits TTX-sensitive neuronal subtypes, slightly inhibits skeletal muscle $Na_V1.4$ and cardiac $Na_V1.5$, and shows no effect on TTX-resistant neuronal subtypes (Xiao et al., 2008). HWTX-IV reportedly inhibits $Na_V1.2$ ($IC_{50}$ ∼150 nM), $Na_V1.3$ ($IC_{50}$ ∼340 nM) and $Na_V1.7$ ($IC_{50}$ ∼17, 22.7 and 26 nM) more strongly than $Na_V1.4$ and 1.5 with $IC_{50}$ values >10 $\mu$M (Revell et al., 2013; Xiao et al., 2008, 2010). In this study, we recorded an $IC_{50}$ for HWTX-IV of 14.6 nM, a value close to that recorded for $Na_V1.7$ channels in previous studies and lower than the value of 30 nM, which characterises these VGSCs as $Na_V1.7$.

To further confirm the subtype pharmacology, we used ProTx-II, a *Thrixopelma pruriens* tarantula neurotoxin reported to inhibit multiple VGSC subtypes ($Na_V1.1$–1.8) but reported to be the most potent blocker of $Na_V1.7$ to date, with an approximately100-fold selectivity over other subtypes (Schmalhofer et al., 2008; Smith et al., 2007). ProTx-II acts on neurotoxin site 4 by binding to the DIIS3-S4 linker and hence trapping the DIIS4 linker in the resting state (Park et al., 2014). ProTx-II has been shown to block $Na_V1.2$, $Na_V1.4$ and $Na_V1.6$ with $IC_{50}$ values of 41, 39 and 26 nM, respectively, while an almost complete blockage of $Na_V1.7$ currents was achieved with 10 nM and $IC_{50}$ values of 0.3 and 0.7 nM recorded, respectively (Schmalhofer et al., 2008; Xiao et al., 2010). Similar results were obtained in our study as the TE671 cell sodium currents were suppressed by about 85% following application of 10 nM ProTx-II and with an $IC_{50}$ value of 0.8 nM (Fig. 6). As this value correlates tightly with the previously reported $IC_{50}$ values for $Na_V1.7$, it serves as a strong indication that the pharmacology of the VGSC expressed by TE671 cells is that of the $Na_V1.7$ subtype. ProTx-II selectivity for $Na_V1.7$ is achieved due to the conserved amino acid residues at the DIIS3−S4 extracellular loop, different from all other $Na_V$ subtypes (de Lera Ruiz & Kraus, 2015).

Though one would expect a rhabdomyosarcoma skeletal muscle cell line to inherently express the skeletal muscle $Na_V1.4$ channel, our results tell a different story where the sodium currents are predominantly mediated by the $Na_V1.7$ channel. However, there is now resounding evidence that beyond the native regional expression of the various VGSC subtypes in neurons, cardiomyocytes, myocytes and other excitable cells, it is also possible to express a repertoire of VGSCs playing functional roles in cells that are otherwise considered as non-excitable. Various VGSC subtypes are known to be functionally expressed in cells where they may perform functions other than electrogenesis, for example, the expression of both $Na_V1.4$ and $Na_V1.7$ in red blood cells (Hoffman et al., 2004); $Na_V1.2$, $Na_V1.3$, $Na_V1.6$, $Na_V1.7$ in cardiac fibroblasts (Li et al., 2009); and TTX sensitive channels and $Na_V1.5$ in T-lymphocytes (DeCoursey et al., 1985; Lo et al., 2012). Similarly, substantial evidence now exists that VGSCs are found in cancerous cells where they are thought to perform many non-canonical functions and can be pursued as therapeutic targets. The involvement of VGSCs in human cancer invasion and growth has been well documented; for example while $Na_V1.5$ has been attributed to the migration and invasion in metastatic breast cancer (Yang et al., 2012), $Na_V1.7$ is now known to be a novel marker for human prostate cancer (Brackenbury & Djamgoz, 2006; Diss et al., 2005) and VGSCs may play a role in the metastatic behaviour of cancer cells (Patel & Brackenbury, 2015). As TE671 is a cancerous cell line, the expression of $Na_V1.7$ channels in this muscle cell line may be related to its origins.

There has been a presumption that the TE671 rhabdomyosarcoma cell line in its undifferentiated form expresses the skeletal muscle $Na_V1.4$ channel; however, our results demonstrate that it is the $Na_V1.7$ subtype RNA that is predominately transcribed in the TE671 cell line. We also provide pharmacological evidence that the expressed channels are predominantly the $Na_V1.7$ subtype. As $Na_V1.7$ is a major target for the study of pain and for modern analgesic drug discovery, the TE671 cell line provides a very convenient and cost-effective model for research in this field. The use of this cell line also constitutes a refinement, which could reduce the need for animal use in studies of pain.

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

## Additional information

### Data availability statement

All data are presented in the manuscript or the Supplementary information. Data are available upon request to the corresponding author.

### Competing interests

The authors declare that the research was conducted in the absence of any commercial or financial relationships that could be construed as a potential conflict of interest.

### Author contributions

All authors contributed to: conception or design of the work; acquisition or analysis or interpretation of data for the work; drafting the work or revising it critically for important intellectual content. All authors have read and approved the final version of this manuscript and agree to be accountable for all aspects of the work in ensuring that questions related to the accuracy or integrity of any part of the work are appropriately investigated and resolved. All persons designated as authors qualify for authorship, and all those who qualify for authorship are listed.

### Funding

This work was funded by a PhD Scholarships from the UK Commonwealth Scholarship Commission (CMCS-2014-114) to N.N. and from the Ministry of Education, Malaysia (KPT(BS)840916035943) to M.A.

### Authors' present addresses

N. M. Ngum: Aston University, Birmingham, B4 7ET, UK.

M. Y. A. Aziz: UniSZA Science and Medicine Foundation Centre, Universiti Sultan Zainal Abidin, Gong Badak Campus, 21 300 Kuala Nerus, Terengganu, Malaysia.

R. J. Wall: Division of Biological Chemistry and Drug Discovery, Wellcome Centre for Anti-Infectives Research, School of Life Sciences, University of Dundee, Dundee DD1 5EH, UK.

### Keywords

Huwentoxin-IV, Na$_V$1.7, patch-clamp, Protoxin-II, TE671 cells, tetrodotoxin, voltage-gated sodium channel

## Supporting information

Additional supporting information can be found online in the Supporting Information section at the end of the HTML view of the article. Supporting information files available:

**Peer Review History**
**Statistical Summary Document**

