## [Peer Review History · The Journal of Physiology]

Non-canonical endogenous expression of voltage-gated sodium channel NaV1.7 subtype by the TE671 rhabdomyosarcoma cell line

Neville M Ngum, Muhammad Y A Aziz, Muhammad Liaque Latif, Richard J Wall, Ian R Duce, and Ian R Mellor
DOI: 10.1113/JP283055

Corresponding author(s): Ian Mellor (ian.mellor@nottingham.ac.uk)

The following individual(s) involved in review of this submission have agreed to reveal their identity: Rajesh Khanna (Referee #1); Mohammed A Nassar (Referee #2)

Review Timeline:

Submission Date:	29-Jun-2021
Editorial Decision:	21-Jul-2021
Revision Received:	18-Aug-2021
Editorial Decision:	02-Sep-2021
Resubmission Received:	07-Mar-2022
Editorial Decision:	22-Mar-2022
Revision Received:	24-Mar-2022
Accepted:	05-Apr-2022

Senior Editor: Ian Forsythe

Reviewing Editor: Nikita Gamper

Transaction Report:

Dear Dr Mellor,

Re: JP-RP-2021-282097 "Non-canonical endogenous expression of voltage-gated sodium channel NaV1.7 subtype by the TE671 rhabdomyosarcoma cell line" by Neville M Ngum, Muhammad Y A Aziz, Richard J Wall, Ian R Duce, and Ian R Mellor

Thank you for submitting your manuscript to The Journal of Physiology. It has been assessed by a Reviewing Editor and by 2 expert Referees and I am pleased to tell you that it is considered to be acceptable for publication following satisfactory revision.

The reports are copied at the end of this email. Please address all of the points and incorporate all requested revisions, or explain in your Response to Referees why a change has not been made.

NEW POLICY: In order to improve the transparency of its peer review process The Journal of Physiology publishes online as supporting information the peer review history of all articles accepted for publication. Readers will have access to decision letters, including all Editors' comments and referee reports, for each version of the manuscript and any author responses to peer review comments. Referees can decide whether or not they wish to be named on the peer review history document.

I hope you will find the comments helpful and have no difficulty returning your revisions within 4 weeks.

Your revised manuscript should be submitted online using the links in Author Tasks Link Not Available.

Any image files uploaded with the previous version are retained on the system. Please ensure you replace or remove all files that have been revised.

REVISION CHECKLIST:

- Article file, including any tables and figure legends, must be in an editable format (eg Word)
- Upload each figure as a separate high quality file
- Upload a full Response to Referees, including a response to any Senior and Reviewing Editor Comments;
- Upload a copy of the manuscript with the changes highlighted.

- A potential 'Cover Art' file for consideration as the Issue's cover image;
- Appropriate Supporting Information (Video, audio or data set https://jp.msubmit.net/cgi-bin/main.plex?form_type=display_requirements#supp).

To create your 'Response to Referees' copy all the reports, including any comments from the Senior and Reviewing Editors, into a Word, or similar, file and respond to each point in colour or CAPITALS and upload this when you submit your revision.

I look forward to receiving your revised submission.

If you have any queries please reply to this email and staff will be happy to assist.

Yours sincerely,

Ian D. Forsythe
Deputy Editor-in-Chief
The Journal of Physiology
<https://jp.msubmit.net>
<http://jp.physoc.org>
The Physiological Society
Hodgkin Huxley House
30 Farringdon Lane

London, EC1R 3AW
UK
<http://www.physoc.org>
<http://journals.physoc.org>

EDITOR COMMENTS

Reviewing Editor:

Comments for Authors to ensure the paper complies with the Statistics Policy:

In preparing your revision please refer to the Journal's 'Statistics Policy' guidelines:

https://jp.msubmit.net/cgi-bin/main.plex?form_type=display_requirements#statistics

If $n < 30$, all data points must be plotted in the figure in a way that reveals their range and distribution. A bar graph with data points overlaid, a box and whisker plot or a violin plot (the latter two also preferably with data points included) are acceptable formats. SD should be used instead of SEM; all 'n' values must be clearly stated in main text, figures and their legends. The exact p values should be provided for statistical tests. A Statistical Summary document must be provided with the revised manuscript (see Statistics Policy for a template).

Comments to the Author:

The manuscript reports robust expression of nociceptive-neuron-specific voltage gated Na⁺ channel subunit Nav1.7 in the TE671 rhabdomyosarcoma cell line. This is an important finding as Nav1.7 is a verified drug target for pain and cellular models for pharmacological screenings against Nav1.7 are highly sought after. The PCR and pharmacology data are convincing and the manuscript is well-written. The reviewers are broadly supportive of the study, however, reviewer #2 identified several aspects in which the study should be improved. These include verifying Nav1.7 protein expression as well as testing small-molecule Nav1.7 inhibitors. Indeed, as the functional expression of Nav1.7 in TE671 cells is the main finding of this study, these additional tests are necessary. The reviewer also asked to clarify the PCR data.

Senior Editor:

Comments for Authors to ensure the paper complies with the Statistics Policy:
See comments from RE

REFEREE COMMENTS

Referee #1:

The authors document the 'rogue' expression of voltage-gated sodium currents in the TE671 rhabdomyosarcoma skeletal muscle cell line. They present convincing data supporting the transcription of Nav1.7 mRNA and follow that up with pharmacological evidence that the expressed channels are primarily the Nav1.7 subtype. The short manuscript is well written, the results are presented in a clear and logical manner and the conclusions are supported by their data.

Referee #2:

Ngum et al examined voltage activated sodium currents in TE671 using molecular and physiological methods to determine the expressed subtypes. PCR and qPCR showed that the level of mRNA for Nav1.7 is much higher than all other subtypes. The VGS current was blocked with tetrodotoxin, Huwentoxin-IV and Protoxin-II. The authors suggest that this makes TE671 a useful system for the development of analgesic drugs.

In general, the manuscript is well written and flows very well. This applies to all sections which is commendable and appreciated. However, I have few points for the authors to address

1) In figure 1, it seems that the intensity of Nav1.7 bands is greater the higher the passage number is (relative to the GAPDH). Can the authors comments on this please? I suggest they add a figure comparing the level of Nav1.7 in different passages.

2) I am not sure how to interpret the fold difference between the expression of Nav1.7 and 1.4 in 1d in light of the lack of any product in 1a-c! In 1d there are differences between the levels for the channels that are "negative" in 1a-c. My questions is, are you saying the other subtypes are functionally expressed but 1.7 is expressed at a 200 fold higher level, OR are saying there is no evidence of biologically relevant expression of all subtypes except 1.7? Should a ratio of GAPDH of 0.0001 or lower be presented as Zero expression? A 200 fold difference sounds impressive but I feel it is not meaningful.

3) I expected an attempt to detect the 1.7 protein by either ICC or a western blot. I am not sure why this is not included in the molecular characterisation. There are good antibodies for 1.7 (e.g. from Neuromab) that can be used to support the qPCR data.

4) I wonder why one of the published small molecular inhibitors of Nav1.7 (e.g. PF-05089771) was not included in your study. Can you please comment on this?

END OF COMMENTS

Confidential Review

29-Jun-2021

JP-RP-2021-282097 "Non-canonical endogenous expression of voltage-gated sodium channel Nav1.7 subtype by the TE671 rhabdomyosarcoma cell line" by Neville M Ngum, Muhammad Y A Aziz, Richard J Wall, Ian R Duce, and Ian R Mellor

We should like to thank the editors and the referees for taking the time to review our manuscript and for their constructive comments on it. Please find our responses below.

RESPONSES TO EDITOR AND REFEREES

EDITOR COMMENTS

Reviewing Editor:

Comments for Authors to ensure the paper complies with the Statistics Policy:

In preparing your revision please refer to the Journal's 'Statistics Policy' guidelines:

If $n < 30$, all data points must be plotted in the figure in a way that reveals their range and distribution. A bar graph with data points overlaid, a box and whisker plot or a violin plot (the latter two also preferably with data points included) are acceptable formats. SD should be used instead of SEM; all 'n' values must be clearly stated in main text, figures and their legends. The exact p values should be provided for statistical tests. A Statistical Summary document must be provided with the revised manuscript (see Statistics Policy for a template).

Response: We have added individual data points to the bar graph in Figure 1D. We have ensured that all mean values quoted in the text are accompanied by SD and n, and that P has been given for any statistical comparisons. A statistical summary has been completed and submitted.

Comments to the Author:

The manuscript reports robust expression of nociceptive-neuron-specific voltage gated Na⁺ channel subunit Nav1.7 in the TE671 rhabdomyosarcoma cell line. This is an important finding as Nav1.7 is a verified drug target for pain and cellular models for pharmacological screenings against Nav1.7 are highly sought after. The PCR and pharmacology data are convincing and the manuscript is well-written. The reviewers are broadly supportive of the study, however, reviewer #2 identified several aspects in which the study should be improved. These include verifying Nav1.7 protein expression as well as testing small-molecule Nav1.7 inhibitors. Indeed, as the functional expression of Nav1.7 in TE671 cells is the main finding of this study, these additional tests are necessary. The reviewer also asked to clarify the PCR data.

Response:

It would be impossible to conduct the two additional sets of experiments that have been suggested within a response time of only four weeks and it would even be challenging to obtain the reagents in the current climate. However, both of these suggestions were considered during our study and we felt that neither of these provided additional weight to the case. We have given our reasons for this in our response to Referee #2 below.

REFEREE COMMENTS

Referee #1:

The authors document the 'rogue' expression of voltage-gated sodium currents in the TE671 rhabdomyosarcoma skeletal muscle cell line. They present convincing data supporting the transcription of Nav1.7 mRNA and follow that up with pharmacological evidence that the expressed channels are primarily the Nav1.7 subtype. The short manuscript is well written, the results are presented in a clear and logical manner and the conclusions are supported by their data.

Response:

There are no points to respond to but we thank Referee #1 for their positive comments about our manuscript.

Referee #2:

Ngum et al examined voltage activated sodium currents in TE671 using molecular and physiological methods to determine the expressed subtypes. PCR and qPCR showed that the level of mRNA for Nav1.7 is much higher than all other subtypes. The VGS current was blocked with tetrodotoxin, Huwentoxin-IV and Protoxin-II. The authors suggest that this makes TE671 a useful system for the development of analgesic drugs.

In general, the manuscript is well written and flows very well. This applies to all sections which is commendable and appreciated. However, I have few points for the authors to address

1) In figure 1, it seems that the intensity of Nav1.7 bands is greater the higher the passage number is (relative to the GAPDH). Can the authors comments on this please? I suggest they add a figure comparing the level of Nav1.7 in different passages.

Response: We have added a comment to the main text on page 9 (highlighted in red) noting that "There was also an apparent increase in band intensity with increasing passage number", however, because the qRT-PCR was only conducted for passage 10 we are unable to precisely quantify the apparent increase in expression.

2) I am not sure how to interpret the fold difference between the expression of Nav1.7 and 1.4 in 1d in light of the lack of any product in 1a-c! In 1d there are differences between the levels for the channels that are "negative" in 1a-c. My questions is, are you saying the other subtypes are functionally expressed but 1.7 is expressed at a 200 fold higher level, OR are saying there is no evidence of biologically relevant expression of all subtypes except 1.7? Should a ratio of GAPDH of 0.0001 or lower be presented as Zero expression? A 200 fold difference sounds impressive but I feel it is not meaningful.

The data presented in Figure 1D are derived from the qRT-PCR experiments that provide much more sensitive detection of RNA levels than visualising the PCR products on the gels. Perhaps the log scale of the plot in Figure 1D may have been misleading and showing up levels and differences that were effectively negligible and, therefore, not visible on the gels. Therefore, we have changed the Y-axis in this plot back to a linear scale. The >200 fold difference quoted in the text is based on the qRT-PCR data comparing Nav_v1.7 to Nav_v1.4, which is the next highest in terms of expression. In answer to the referee's question, we are saying that Nav_v1.7 is expressed at a > 200 fold higher level than Nav_v1.4 but also it may be true that the very low level of Nav_v1.4 expression (and lower for others) is not biologically relevant. On page 9 we have clarified the text a little and added "In effect, the only biologically relevant level of RNA is of that coding for Nav_v1.7" (highlighted in red). We hope this clarifies the situation.

3) I expected an attempt to detect the 1.7 protein by either ICC or a western blot. I am not sure why this is not included in the molecular characterisation. There are good antibodies for 1.7 (e.g. from Neuromab) that can be used to support the qPCR data.

Response: Interestingly, we considered this. However, we felt that ICC or Western blot would only be meaningful if done with the full set of Nav antibodies and, in any case, this would only show that we have Nav_v1.7 protein; we have gone one step further and show functional membrane protein with clear Nav_v1.7 pharmacology. This is well supported by the clear dominance of Nav_v1.7 RNA. So we didn't feel that this further experiment added anything useful to the study.

4) I wonder why one of the published small molecular inhibitors of Nav1.7 (e.g. PF-05089771) was not included in your study. Can you please comment on this?

Response: We compiled a list of candidate inhibitors with some level of Nav_v1.7 selectivity for the pharmacological study. PF-05089771 was considered as one these candidates but rejected on the grounds that it is not as selective as ProTx-II or HWTX-IV; only about 10x over Nav_v1.2 and 1.6, and there is limited data for Nav_v1.8 and 1.9, some of which suggests that it is selective for Nav_v1.8 also.

Dear Dr Mellor,

Re: JP-RP-2021-282097R1 "Non-canonical endogenous expression of voltage-gated sodium channel NaV1.7 subtype by the TE671 rhabdomyosarcoma cell line" by Neville M Ngum, Muhammad Y A Aziz, Richard J Wall, Ian R Duce, and Ian R Mellor

Thank you for submitting your manuscript to The Journal of Physiology. It has been assessed by a Reviewing Editor and by 1 Referees and the reports are copied below.

Please let your co-authors know of the following editorial decision as quickly as possible.

As you will see, in its current form, the manuscript is not acceptable for publication in The Journal of Physiology. In comments to me, the Reviewing Editor expressed interest in the potential of this study, but much work still needs to be done (and this may include new experiments) in order to satisfactorily address the concerns raised in the reports.

In view of this interest, I would like to offer you the opportunity to carry out all of the changes requested in full, and to resubmit a new manuscript using the "Submit Special Case Resubmission for JP-RP-2021-282097R1..." on your homepage.

We cannot, of course, guarantee ultimate acceptance at this stage as the revisions required are substantial. However, we encourage you to consider the requested changes and resubmit your work to us if you are able to complete or address all changes.

A new manuscript would be renumbered and redated, but the original referees would be consulted wherever possible. An additional referee's opinion could be sought, if the Reviewing Editor felt it necessary. A full response to each of the reports should be uploaded with a new version.

I hope that the points raised in the reports will be helpful to you.

Yours sincerely,

Ian D. Forsythe
Deputy Editor-in-Chief
The Journal of Physiology
<https://jp.msubmit.net>
<http://jp.physoc.org>
The Physiological Society
Hodgkin Huxley House
30 Farringdon Lane
London, EC1R 3AW
UK
<http://www.physoc.org>
<http://journals.physoc.org>

EDITOR COMMENTS

Reviewing Editor:

Thank you for your revision. In moving a manuscript from provisional acceptance to final acceptance the Editors take particular note of whether specific requests have been completed. In this case two relatively straightforward additional experiments were requested; without this information the manuscript is of lesser impact and would not likely reach our criterion for acceptance. If the additional experiments will take more time, then we are happy to provide extra time (in this case we would technically reject this version but invite you to re-submit, as a special case resubmission).

Senior Editor:

Comments for Authors to ensure the paper complies with the Statistics Policy:
Dose response curves need raw data points.

Comments to the Author:

I agree with the Editor's assessment that this MS needs to show the additional data in order proceed to publication. We are happy to give you more time to do this.

REFeree COMMENTS

Referee #2:

I understand and accept that an ICC is logistically not possible to set up in 4 weeks.

The change to result section 3.1 and the relevant figure deals well with my point.

I am clear now on the justification for using the protein blockers, please include this in the manuscript.

ADDITIONAL FORMATTING REQUIREMENTS:

Please could authors include:

- * Author contributions
- * Acknowledgements

END OF COMMENTS

JP-RP-2021-282097 "Non-canonical endogenous expression of voltage-gated sodium channel NaV1.7 subtype by the TE671 rhabdomyosarcoma cell line" by Neville M Ngum, Muhammad Y A Aziz, M Liaque Latif, Richard J Wall, Ian R Duce, and Ian R Mellor

RESPONSES TO EDITORS AND REFEREE #2

EDITOR COMMENTS

Reviewing Editor:

Thank you for your revision. In moving a manuscript from provisional acceptance to final acceptance the Editors take particular note of whether specific requests have been completed. In this case two relatively straightforward additional experiments were requested; without this information the manuscript is of lesser impact and would not likely reach our criterion for acceptance. If the additional experiments will take more time, then we are happy to provide extra time (in this case we would technically reject this version but invite you to re-submit, as a special case resubmission).

Senior Editor:

Comments for Authors to ensure the paper complies with the Statistics Policy:
Dose response curves need raw data points.

Comments to the Author:

I agree with the Editor's assessment that this MS needs to show the additional data in order proceed to publication. We are happy to give you more time to do this.

RESPONSE:

We thank you for the provision of more time to complete further experimental work and the opportunity to resubmit this. We have now performed a Western Blot according to the original request of Referee #2 and have added this to the manuscript (text and new Figure 2). An additional author has been added, who performed this work. We have also added raw data points to the dose-response curves in Figures 4-6 as requested. A highlighted copy of the manuscript has been provided indicating all of the additions.

REFEREE COMMENTS

Referee #2:

I understand and accept that an ICC is logistically not possible to set up in 4 weeks.

RESPONSE: We were given additional time and have now performed a Western Blot.

The change to result section 3.1 and the relevant figure deals well with my point.

RESPONSE: Thank you.

I am clear now on the justification for using the protein blockers, please include this in the manuscript.

RESPONSE: We added a note in our methods (section 2.5) justifying the selection of the peptide spider toxins over the small molecule inhibitors like PF-05089771 on the grounds of their much superior and well documented selectivity for Na_v1.7.

ADDITIONAL FORMATTING REQUIREMENTS:

Please could authors include:

- * Author contributions
- * Acknowledgements

RESPONSE:

We have now included the author contributions within the manuscript file. We have no further acknowledgements beyond the funding bodies already given.

Dear Dr Mellor,

Re: JP-RP-2022-283055X "Non-canonical endogenous expression of voltage-gated sodium channel NaV1.7 subtype by the TE671 rhabdomyosarcoma cell line" by Neville M Ngum, Muhammad Y A Aziz, Muhammad Liaque Latif, Richard J Wall, Ian R Duce, and Ian R Mellor

Thank you for submitting your revised Research Article to The Journal of Physiology. It has been assessed by the original Reviewing Editor and Referees and has been well received. Some final revisions have been requested.

The reports are copied at the end of this email. Please address all of the points and incorporate all requested revisions, or explain in your Response to Referees why a change has not been made.

NEW POLICY: In order to improve the transparency of its peer review process The Journal of Physiology publishes online as supporting information the peer review history of all articles accepted for publication. Readers will have access to decision letters, including all Editors' comments and referee reports, for each version of the manuscript and any author responses to peer review comments. Referees can decide whether or not they wish to be named on the peer review history document.

Authors are asked to use The Journal's premium BioRender (<https://biorender.com/>) account to create/redraw their Abstract Figures. Information on how to access The Journal's premium BioRender account is here: <https://physoc.onlinelibrary.wiley.com/journal/14697793/biorender-access> and authors are expected to use this service. This will enable Authors to download high-resolution versions of their figures. The link provided should only be used for the purposes of this submission. Authors will be charged for figures created on this premium BioRender account if they are not related to this manuscript submission.

I hope you will find the comments helpful and have no difficulty returning your revisions within 2 weeks.

Your revised manuscript should be submitted online using the links in Author Tasks Link Not Available.

Any image files uploaded with the previous version are retained on the system. Please ensure you replace or remove all files that have been revised.

REVISION CHECKLIST:

- Article file, including any tables and figure legends, must be in an editable format (eg Word)
- Abstract figure file (see above)
- Statistical Summary Document
- Upload each figure as a separate high quality file
- Upload a full Response to Referees, including a response to any Senior and Reviewing Editor Comments;
- Upload a copy of the manuscript with the changes highlighted.

- A potential 'Cover Art' file for consideration as the Issue's cover image;
- Appropriate Supporting Information (Video, audio or data set https://jp.msubmit.net/cgi-bin/main.plex?form_type=display_requirements#supp).

To create your 'Response to Referees' copy all the reports, including any comments from the Senior and Reviewing Editors, into a Word, or similar, file and respond to each point in colour or CAPITALS and upload this when you submit your revision.

I look forward to receiving your revised submission.

If you have any queries please reply to this email and staff will be happy to assist.

Yours sincerely,

Ian D. Forsythe
Deputy Editor-in-Chief
The Journal of Physiology
<https://jp.msubmit.net>
<http://jp.physoc.org>
The Physiological Society
Hodgkin Huxley House
30 Farringdon Lane
London, EC1R 3AW
UK
<http://www.physoc.org>
<http://journals.physoc.org>

REQUIRED ITEMS:

-You must upload original, uncropped western blot/gel images (including controls) if they are not included in the manuscript. This is to confirm that no inappropriate, unethical or misleading image manipulation has occurred
<https://physoc.onlinelibrary.wiley.com/hub/journal-policies#imagmanip> These should be uploaded as 'Supporting information for review process only'. Please label/highlight the original gels so that we can clearly see which sections/lanes have been used in the manuscript figures.

-Papers must comply with the Statistics Policy https://jp.msubmit.net/cgi-bin/main.plex?form_type=display_requirements#statistics

In summary:

-If $n \leq 30$, all data points must be plotted in the figure in a way that reveals their range and distribution. A bar graph with data points overlaid, a box and whisker plot or a violin plot (preferably with data points included) are acceptable formats.

-If $n > 30$, then the entire raw dataset must be made available either as supporting information, or hosted on a not-for-profit repository e.g. FigShare, with access details provided in the manuscript.

- 'n' clearly defined (e.g. x cells from y slices in z animals) in the Methods. Authors should be mindful of pseudoreplication.

-All relevant 'n' values must be clearly stated in the main text, figures and tables, and the Statistical Summary Document (required upon revision)

-The most appropriate summary statistic (e.g. mean or median and standard deviation) must be used. Standard Error of the Mean (SEM) alone is not permitted.

-Exact p values must be stated. Authors must not use 'greater than' or 'less than'. Exact p values must be stated to three significant figures even when 'no statistical significance' is claimed.

-Statistics Summary Document completed appropriately upon revision

EDITOR COMMENTS

Reviewing Editor:

Comments for Authors to ensure the paper complies with the Statistics Policy:

It seems that Figure 3 does not include primary data and it also uses S.E.M. rather than SD, which is against journal's policy.

If the Statistical Summary Document has errors please describe what is incorrect? :

Several panels of Fig. 3C report mean data but only the data for panel 3C is included in the statistical summary.

Comments to the Author:

The revision adequately addressed reviewer's concerns. There are still few minor inconsistencies with regards to data presentation and statistics (i.e. Fig. 3 may not be in full compliance with the journal style).

Senior Editor:

Comments to the Author:

The authors have done a good job in providing the requested Western blot data. There are just a few minor issues to finish

off the MS, most of which relate to the SSD. Please read through this carefully and make the SSD is as complete and informative as possible. Look at a few other published examples.

I note that the authors should just cite the existing literature demonstrating specificity of the Nav1.7 antibody where others have previously used this Ab; there is not need to provide the further controls requested. e.g. J Pain. 2021 Jul; 22(7): 806-816.

The authors may choose to keep the Figure 3 presentation as is (rather than add all the raw data points as requested by one referee).

The authors may care to note in the discussion that the use of this cell line would constitute a refinement which could reduce the need for animal use in studies of pain (but the authors may choose to ignore this comment).

REFEREE COMMENTS

Referee #1:

The authors have satisfactorily addressed the points raised. However, in figures four through six, the kinetics should be presented, please provide the activation and inactivation curves.

Referee #2:

The authors addressed the points raised in the reviewers. Specifically the addition of a western blot.

However, the western blot data is not convincing on their own because of the way the experiment was done. They are only fine considering the mRNA data. The issues I noted with the WB are

1) The antibody used (Nav1.7 Antibody #14573) is recommended for mouse and rat channels (as indicated by the supplier's website).

2) The gel does not have a positive control (primary DRG, or a cell line known to express 1.7 like N2a).

2) The gel does not have a negative control (brain cortex or any other cell line)!!!

However, with the mRNA and physiology, they collectively support the authors' conclusion.

END OF COMMENTS

2nd Confidential Review

07-Mar-2022

JP-RP-2021-282097 "Non-canonical endogenous expression of voltage-gated sodium channel NaV1.7 subtype by the TE671 rhabdomyosarcoma cell line" by Neville M Ngum, Muhammad Y A Aziz, M Liaque Latif, Richard J Wall, Ian R Duce, and Ian R Mellor

RESPONSES TO EDITORS AND REFEREES

We thank the Editors and Referees again for their time and effort spent on our manuscript and their useful comments.

EDITOR COMMENTS

Reviewing Editor:

Comments for Authors to ensure the paper complies with the Statistics Policy:
It seems that Figure 3 does not include primary data and it also uses S.E.M. rather than SD, which is against journal's policy.

Response: Because of the relatively high number of points, the larger SD error bars detracted from the relationships described by the data. Please see comment below from Senior Editor too.

If the Statistical Summary Document has errors please describe what is incorrect? :
Several panels of Fig. 3C report mean data but only the data for panel 3C is included in the statistical summary.

Response: All data from Figure 3 is now included in the Statistical Summary Document.

Comments to the Author:

The revision adequately addressed reviewer's concerns. There are still few minor inconsistencies with regards to data presentation and statistics (i.e. Fig. 3 may not be in full compliance with the journal style).

Response: See responses above.

Senior Editor:

Comments to the Author:

The authors have done a good job in providing the requested Western blot data. There are just a few minor issues to finish off the MS, most of which relate to the SSD. Please read through this carefully and make the SSD is as complete and informative as possible. Look at a few other published examples.

Response: We have now included all of the data from Figure 3 in the SSD.

I note that the authors should just cite the existing literature demonstrating specificity of the Nav1.7 antibody where others have previously used this Ab; there is not need to provide the further controls requested. e.g. J Pain. 2021 Jul; 22(7): 806-816.

Response: We have cited the suggested reference in the methods section.

The authors may choose to keep the Figure 3 presentation as is (rather than add all the raw data points as requested by one referee).

Response: We thank you for this. Adding the raw data points would have added and extra 350 points to Figure 3B, a further 955 points to Figure 3D and a further 2550 points to Figure 3E. This was very detrimental to the presentation.

The authors may care to note in the discussion that the use of this cell line would constitute a refinement which could reduce the need for animal use in studies of pain (but the authors may choose to ignore this comment).

Response: We like this comment and have added it to our closing remarks with very similar wording (hope that is OK).

REFEREE COMMENTS

Referee #1:

The authors have satisfactorily addressed the points raised. However, in figures four through six, the kinetics should be presented, please provide the activation and inactivation curves.

Response: Those experiments were conducted using repetitive depolarisations from -80 mV to a consistent test potential of -10 mV at 5 s intervals to observe the extent and time course of inhibition. So the activation and inactivation curves are not available.

Referee #2:

The authors addressed the points raised in the reviewers. Specifically the addition of a western blot.

However, the western blot data is not convincing on their own because of the way the experiment was done. They are only fine considering the mRNA data. The issues I noted with the WB are

- 1) The antibody used (Nav1.7 Antibody #14573) is recommended for mouse and rat channels (as indicated by the supplier's website).
- 2) The gel does not have a positive control (primary DRG, or a cell line known to express 1.7 like N2a).
- 2) The gel does not have a negative control (brain cortex or any other cell line)!!!

However, with the mRNA and physiology, they collectively support the authors' conclusion.

Response: Please see the comment from the Senior Editor.

Dear Dr Mellor,

Re: JP-RP-2022-283055XR1 "Non-canonical endogenous expression of voltage-gated sodium channel NaV1.7 subtype by the TE671 rhabdomyosarcoma cell line" by Neville M Ngum, Muhammad Y A Aziz, Muhammad Liaque Latif, Richard J Wall, Ian R Duce, and Ian R Mellor

I am pleased to tell you that your paper has been accepted for publication in The Journal of Physiology.

NEW POLICY: In order to improve the transparency of its peer review process The Journal of Physiology publishes online as supporting information the peer review history of all articles accepted for publication. Readers will have access to decision letters, including all Editors' comments and referee reports, for each version of the manuscript and any author responses to peer review comments. Referees can decide whether or not they wish to be named on the peer review history document.

The last Word version of the paper submitted will be used by the Production Editors to prepare your proof. When this is ready you will receive an email containing a link to Wiley's Online Proofing System. The proof should be checked and corrected as quickly as possible.

Authors should note that it is too late at this point to offer corrections prior to proofing. The accepted version will be published online, ahead of the copy edited and typeset version being made available. Major corrections at proof stage, such as changes to figures, will be referred to the Reviewing Editor for approval before they can be incorporated. Only minor changes, such as to style and consistency, should be made a proof stage. Changes that need to be made after proof stage will usually require a formal correction notice.

All queries at proof stage should be sent to TJP@wiley.com

Are you on Twitter? Once your paper is online, why not share your achievement with your followers. Please tag The Journal (@jphysiol) in any tweets and we will share your accepted paper with our 23,000+ followers!

Yours sincerely,

Ian D. Forsythe
Deputy Editor-in-Chief
The Journal of Physiology
<https://jp.msubmit.net>
<http://jp.physoc.org>
The Physiological Society
Hodgkin Huxley House
30 Farringdon Lane
London, EC1R 3AW
UK
<http://www.physoc.org>
<http://journals.physoc.org>

P.S. - You can help your research get the attention it deserves! Check out Wiley's free Promotion Guide for best-practice recommendations for promoting your work at www.wileyauthors.com/eeo/guide. And learn more about Wiley Editing Services which offers professional video, design, and writing services to create shareable video abstracts, infographics, conference posters, lay summaries, and research news stories for your research at www.wileyauthors.com/eeo/promotion.

*** IMPORTANT NOTICE ABOUT OPEN ACCESS ***

Information about Open Access policies can be found here <https://physoc.onlinelibrary.wiley.com/hub/access-policies>

To assist authors whose funding agencies mandate public access to published research findings sooner than 12 months after publication The Journal of Physiology allows authors to pay an open access (OA) fee to have their papers made freely available immediately on publication.

You will receive an email from Wiley with details on how to register or log-in to Wiley Authors Services where you will be able to place an OnlineOpen order.

You can check if your funder or institution has a Wiley Open Access Account here <https://authorservices.wiley.com/author-resources/Journal-Authors/licensing-and-open-access/open-access/author-compliance-tool.html>

Your article will be made Open Access upon publication, or as soon as payment is received.

If you wish to put your paper on an OA website such as PMC or UKPMC or your institutional repository within 12 months of publication you must pay the open access fee, which covers the cost of publication.

OnlineOpen articles are deposited in PubMed Central (PMC) and PMC mirror sites. Authors of OnlineOpen articles are permitted to post the final, published PDF of their article on a website, institutional repository, or other free public server, immediately on publication.

Note to NIH-funded authors: The Journal of Physiology is published on PMC 12 months after publication, NIH-funded authors DO NOT NEED to pay to publish and DO NOT NEED to post their accepted papers on PMC.

EDITOR COMMENTS

Reviewing Editor:

I have no further concerns.

Senior Editor:

Thank you for these final amendments.

Congratulations on an interesting Ms.

REFEREE COMMENTS

Referee #1:

The authors have done an excellent job of addressing the concerns.

Referee #2:

I covered these points in my initial review. I have nothing new to add.

3rd Confidential Review

24-Mar-2022